# Transition from 24-Hour Shifts to Safer Work Schedules for Nurses in Latvian Healthcare: Policy Analysis and Recommendations

**DOI:** 10.3390/ijerph22111736

**Published:** 2025-11-17

**Authors:** Olga Cerela-Boltunova, Kristine Klavina

**Affiliations:** 1Department of Nursing and Midwifery, Riga Stradiņš University, LV-1067 Rīga, Latvia; 2Development of Human Resources Sector, Ministry of Health the Republic of Latvia, LV-1067 Rīga, Latvia; kristine.klavina@vm.gov.lv

**Keywords:** work schedule tolerance, nursing staff, health policy, health workforce, Latvia

## Abstract

In Latvia, extended shifts, including 24 h duties, remain common inpatient care settings despite extensive international evidence on their adverse effects on staff well-being and patient safety. We conducted an evidence-informed policy analysis combining a structured review of national legislation and institutional reports with comparative policy mapping across OECD/EU countries. The interpretation was guided by three theoretical frameworks: the Job Demands–Resources model, Effort–Recovery theory, and the Work–Life Interface framework. Latvian practice shows high reliance on long shifts amid workforce shortages and incomplete overtime/rest accounting. In contrast, most OECD and EU countries have implemented 8–12 h multi-shift systems with mandated rest, which are associated with lower error rates, reduced burnout, and higher staff satisfaction. We synthesised four policy options (12 h transition model; 16 h cap; modular 2 × 6 h/3 × 8 h; flexible unit-profiled schedules) and identify seven prerequisites for feasible implementation (regulatory alignment; staffing; financing; management training; digital scheduling; pilot projects; monitoring). A phased transition from 24 h shifts to structured schedules appears both feasible and desirable, with pilot implementation and monitoring aligning with WHO/ILO recommendations. Implications for policy and practice: Reform is a system-level intervention to improve staff well-being, patient safety, and workforce sustainability in Latvia.

## 1. Introduction

In accordance with the Cabinet of Ministers Order No. 1194 (18 December 2024) approving the Health Workforce Development Strategy 2025–2029 [1], an evaluation was initiated to explore flexible models of working time organisation in Latvian healthcare institutions, with a focus on improving staff well-being, particularly in nursing. To ensure conceptual clarity, this paper treats working-time organisation not merely as an administrative issue but as a systemic determinant of occupational health, workforce sustainability, and patient safety.

The quality and sustainability of the healthcare system is largely based on the working conditions of the professionals employed in it, including the organisation of working time. Nurses, as a fundamental element of the healthcare system, face high physical and psycho-emotional stress on a daily basis, which extends beyond individual coping resources when shifts are excessively long, particularly in the case of 24 h duties [2,3,4,5]. Empirical evidence links prolonged work shifts to increased burnout, disturbed work–life balance, and compromised patient safety [6,7,8,9].

In Latvia, the organisation of working time, particularly for nurses, often follows historically established practices rather than evidence-based standards of staff well-being. These practices are maintained due to persistent workforce shortages and economic constraints. According to OECD data, Latvia has one of the lowest nurse densities per 1000 population in the European Union at 4.2 nurses compared to the EU average of over 8.5, creating chronic staff shortages and reinforcing the reliance on extended and 24 h shifts [10].

Such working time arrangements pose significant risks to physical and mental health of the staff, as well as reduce work efficiency and professional sustainability. Studies in Latvia show that more than 60% of nurses regularly work more than 12 h and 38% say they work a 24 h shift at least once a month [11,12]. These data are consistent with international studies confirming that working longer than 12 h increases the likelihood of errors in the care process, contributes to medication administration errors, and reduces patient satisfaction with the care received [13,14,15]. In this analysis, these national figures are not treated as isolated statistics but are interpreted through theoretical models that explain how prolonged working hours function as structural risk factors within the work environment.

The World Health Organization (WHO), the International Labour Organization (ILO), and the European Commission have made clear recommendations on the organisation of working time in healthcare: shifts should not exceed 12 h, there should be at least 11 h of uninterrupted rest between shifts, and the total weekly working time should not exceed 48 h [16,17]. These principles are followed in many EU countries, where the transition to three-shift systems or flexible modular scheduling has already taken place [18,19,20,21,22,23,24,25,26]. This international alignment illustrates a wider policy trend towards protecting the recovery capacity of healthcare staff, an essential mechanism described in the Effort–Recovery theory and Job Demands–Resources (JD-R) model.

In Latvia, these principles are still not fully implemented. 24 h shifts are still common in various healthcare sectors, such as intensive care, emergency services, admissions units, and elsewhere. Work schedules are often drawn up manually or using outdated systems, overtime records are incomplete, and remuneration often does not reflect actual workload [27,28,29]. Long shifts are also maintained for economic reasons by the employees themselves. Night, holiday, and weekend work is paid extra, but in order to secure a monthly income, many are forced to combine several jobs and work up to 60–70 h per week, which significantly exceeds the legal norm [29]. This illustrates a chronic imbalance between job demands and available resources, a core mechanism conceptualised by the JD-R framework.

In this context, the need for a systemic evaluation of working time models has become increasingly relevant. This article presents an evidence-informed policy analysis exploring transition options from 24 h shifts to structured and flexible multi-shift systems in Latvian healthcare. The analysis integrates the Job Demands–Resources (JD-R) model [30], Effort–Recovery theory [31], and Work–Life Interface framework [32], providing a multidimensional lens through which to understand both the organisational and human consequences of prolonged working hours. This theoretical integration also ensures that policy recommendations are anchored not only in empirical data but in established occupational health theory.

## 2. Theoretical Framework

The study is grounded in established occupational psychology and organizational behavior theories that explain the relationship between work demands, recovery, and employee well-being. Integrating these theories provides a conceptual bridge between the empirical evidence on nurse workload and the policy measures required to regulate working time more effectively.

The Job Demands–Resources (JD-R) model [30] provides a conceptual basis for understanding how excessive job demands, such as 24 h shifts, combined with insufficient resources (e.g., staffing, rest, recovery opportunities), lead to burnout, fatigue, and reduced performance. In this context, the duration of shifts represents a structural job demand, while staffing levels, recovery time, and managerial support constitute key resources that can either buffer or exacerbate stress. Within this model, prolonged shift length increases the imbalance between demands and resources, resulting in energy depletion and emotional exhaustion, a phenomenon widely documented in healthcare settings [30,31,32].

The Effort–Recovery theory [33] explains the physiological and cognitive mechanisms underlying recovery. According to this theory, adequate recovery periods support the restoration of functional systems that are taxed during effortful work. When recovery is insufficient, such as in consecutive or extended shifts, physiological and psychological systems remain partially activated, leading to cumulative strain, cognitive impairment, and an elevated risk of burnout. This theoretical lens is particularly relevant for healthcare, where uninterrupted performance under high stress is often valorised but biologically unsustainable.

The Work–Life Interface framework [34] explains how long, unpredictable shifts interfere with personal and family life, reducing overall life satisfaction and increasing turnover intentions. This framework provides an essential complement to the JD-R and Effort–Recovery perspectives, as it captures the broader socio-emotional consequences of irregular work patterns that extend beyond the workplace.

Together, these frameworks offer a multidimensional theoretical foundation for the policy analysis. They highlight how reducing shift length and restructuring scheduling systems can serve as preventive organizational interventions that improve staff well-being, patient safety, and workforce sustainability. The integration of these models allows the analysis to move beyond descriptive policy comparison toward a mechanism-based understanding of how working-time reforms can influence health system outcomes. This theoretical alignment is consistent with WHO (2023) [35] and OECD (2024) [36] recommendations emphasizing safe staffing and the prevention of chronic fatigue among healthcare professionals.

These theoretical perspectives collectively support the policy rationale for regulating working time as an occupational health determinant. They frame fatigue, burnout, and staff turnover not as individual failures but as predictable system-level outcomes of organizational design. This understanding provides a conceptual anchor for the subsequent analysis, where the proposed shift reforms are examined as practical applications of the JD-R and Effort–Recovery mechanisms within Latvia’s healthcare system.

## 3. Policy Analysis Framework and Methods

This study employed a structured and theory-informed policy analysis to evaluate the feasibility of transitioning from 24 h shifts to safer, evidence-based work schedules in Latvian healthcare. Following the logic of the Job Demands–Resources and Effort–Recovery frameworks described above, the analysis conceptualised working-time reform as both a regulatory and psychosocial intervention aimed at restoring balance between job demands and available recovery resources. The methodological design followed the general principles of evidence-informed policy analysis as outlined by the World Health Organization [35] and the OECD [36]. The analytical process unfolded in three complementary stages: contextual mapping of the national situation, comparative review of international policy models, and synthesis of recommendations and feasibility conditions for Latvia.

### 3.1. Data Sources

The study relied on triangulated secondary data derived from legal, institutional, and international evidence bases. This approach ensured both breadth (cross-national comparability) and depth (contextual understanding of Latvia’s system). Data sources included:National legislation and policy documents, such as the Informative Report of the Ministry of Health on Flexible Work Planning Principles, the Cabinet of Ministers’ Orders and Regulations, and the national Health Workforce Strategy [1].National datasets and institutional reports, including the Riga Stradiņš University Workforce Study (2023) [11], data from the Centre for Disease Prevention and Control (SPKC, 2022) [37], and the State Labour Inspectorate’s 2024 report [38]. These were selected based on completeness, transparency, and relevance to healthcare workforce planning.International and comparative sources, including WHO, OECD, and Eurofound databases (2020–2024) [18], as well as peer-reviewed literature on nurse workload, shift scheduling, and burnout [32,39].

Inclusion criteria for international sources required publications issued between 2015 and 2024 in English, reporting quantitative or policy-level findings related to healthcare shift duration, rest regulation, or nurse well-being. The selection strategy ensured that all materials aligned conceptually with the study’s theoretical models by capturing both structural and psychosocial dimensions of working time.

### 3.2. Analytical Process

The policy analysis was conducted through four iterative steps, combining document analysis and comparative evaluation. Each step was reviewed by domain experts to ensure interpretive accuracy:Contextual Mapping: The first stage identified and reviewed the existing Latvian regulatory and institutional framework governing nurses’ work schedules, highlighting areas of inconsistency with EU and WHO standards.Document Review: A systematic review of national and international policy documents, legal frameworks, and academic literature was conducted using targeted keyword searches (“nurse shift system”, “24-h shifts”, “health workforce reform”, “safe staffing”) across PubMed, WHO, and OECD databases. Two independent reviewers conducted relevance screening and cross-validation to minimise selection bias.Comparative Policy Mapping: Synthesis of international experience focusing on countries that have legislatively limited or prohibited 24 h shifts (e.g., Sweden, Norway, Germany, Australia). The analysis compared policy approaches, implementation strategies, and reported outcomes.Synthesis and Feasibility Assessment: Integration of findings to identify enablers, barriers, and preconditions for reform in Latvia, including pilot project feasibility and expected health, safety, and economic effects.

These steps together ensured methodological rigour and provided a multi-level perspective, linking macro-level policy structures with micro-level workforce and well-being determinants, consistent with the theoretical models described in Section 2.

### 3.3. Analytical Framework

To structure the evaluation systematically, the study adopted Walt and Gilson’s Policy Triangle framework [40], which integrates contextual, processual, and actor-oriented dimensions of health policy. This framework was selected for its compatibility with complex, system-level reforms such as working-time restructuring.

Context: Legislative, socio-economic, and workforce conditions in Latvia;Content: Policy alternatives and international models for shift regulation;Process: Mechanisms of policy development, stakeholder involvement, and pilot implementation;Actors: Key institutions responsible for reform (Ministry of Health, hospitals, nursing associations).

By aligning Walt and Gilson’s framework with the JD-R and Effort–Recovery models, this study linked the policy cycle to psychological mechanisms of work stress and recovery—bridging structural and individual levels of analysis.

The analysis did not involve primary data collection. Instead, it relied entirely on publicly available national and international datasets, institutional reports, and policy documents. A purposive sampling strategy was applied to include studies and policies relevant to the Latvian healthcare workforce and comparable OECD/EU contexts. Priority was given to sources demonstrating methodological transparency, data robustness, and direct applicability to the Latvian setting.

Overall, this methodology ensured that the subsequent results (Section 4) emerge from a coherent integration of empirical, policy, and theoretical evidence. It also reinforces that the transition from 24 h shifts is not simply an administrative change but a theoretically grounded occupational health reform.

## 4. Results

### 4.1. Latvia: Current Situation & Compliance

The organisation of working time in the Latvian healthcare system remains heterogeneous and insufficiently structured, with continued reliance on extended and 24 h shifts. This practice remains widespread across several sector, particularly emergency medicine, inpatient care, and intensive care unit, despite extensive international and national evidence of its adverse effects on staff health, cognitive performance, and patient safety [4,5,6,41].

There are currently three main models of shifts in Latvia: (1) the classic three-shift system with 8, 10, or 12 h shifts, (2) a two-shift system, usually in a 12 h format, and (3) a 16 or 24 h shift pattern typical of emergency departments, hospital admissions units, physician assistant work in the regions, and specialised care in critical units. In some cases, 24 h shifts are maintained with employee consent, as they allow concentration of workload into a single shift, longer subsequent rest periods, and additional employment opportunities elsewhere [5,29]. However, within the Job Demands–Resources (JD-R) model, such arrangements clearly represent excessive job demands unmatched by sufficient recovery resources, predisposing staff to strain and exhaustion.

Long-term reliance on extended shifts leads to cumulative fatigue, reduced cognitive performance, and impaired decision-making, effects supported by empirical data. The Riga Stradiņš University workforce study reported that 60% of nurses regularly worked more than 12 h, and 38% at least once per month performed a 24 h shift [11,12]. Moreover, 70% reported fatigue, 44% concentration difficulties, and 28% anxiety or sleep disturbances linked to scheduling. These findings correspond to the Effort–Recovery model’s premise that insufficient rest disrupts physiological and cognitive restoration.

This situation is aggravated by a critical shortage of human resources. In Latvia, there are 4.2 nurses per 1000 population, while the European Union average is over 8.5 nurses per 1000 population [10]. This disproportion means that existing staff have to perform more tasks, work overtime, and often take on double workloads. 24 h shifts are therefore often not a matter of choice, but a practice dictated by necessity in times of resource scarcity.

The 2023 State Labour Inspectorate report [38] identified significant noncompliance with the Labour Law regarding working time, including unrecorded overtime, lack of mandated rest, and incomplete scheduling documentation [27,28]. Consequently, staff work intensively without adequate compensation or recovery, while the bonus-based remuneration system makes longer shifts financially appealing [29]. This feedback loop perpetuates the “high demand–low resource” pattern central to the JD-R model, reinforcing chronic strain and burnout risk.

International standards on working time limits are inconsistently applied in Latvia. WHO and ILO guidelines recommend that shifts not exceed 12 h and include a minimum of 11 h of uninterrupted rest [16,17]. These principles are not explicitly mandated by Latvian legislation, particularly in regional and smaller institutions, leading to inconsistent adherence and potential risks for staff health and patient safety. This compromises staff health and the safety of patient care.

The situation is further complicated by the fact that some staff group, such as midwives, physician assistants, and emergency medical personnel, continue to work extended on-call–style rotations in which relief occurs only the following day. This model persists despite robust evidence of adverse effects on health, sleep quality, and task performance [9,41,42,43,44].

Latvia’s demographic situation and the attraction of young people to the nursing profession further constrain rapid reform: annual graduate numbers only partially cover vacancies, while many early-career nurses leave for the private sector or abroad [45].

Overall, the organisation of working time in Latvia still reflects historical workarounds to staff shortages rather than evidence-based workforce planning. A structured, evidence-informed transition towards flexible, safe, and health-oriented scheduling therefore appears both feasible and necessary to strengthen workforce sustainability and patient safety. Interpreted through the Job Demands–Resources and Effort–Recovery frameworks, these patterns indicate persistently high job demands with inadequate recovery opportunities, conditions associated with burnout and error risk, underscoring the policy need for staged implementation and monitoring. Comparative indicators (Table 1) situate Latvia against EU and Nordic benchmarks, highlighting gaps in nurse staffing density, rest-time compliance, and burnout prevalence.

Comparative indicators highlight Latvia’s divergence from EU and Nordic standards regarding nurse staffing and rest-time compliance (see Table 1).

Building on the contextual overview of current scheduling practices in Latvia, the following section explores how these extended shifts affect healthcare employees’ well-being and professional performance, viewed through the lens of occupational health theories introduced earlier.

### 4.2. Employee Well-Being & Performance

Prolonged work shifts in healthcare associated with substantial adverse effects on staff health, psycho-emotional well-being, and professional performance [3,4,5,6,7,8,9]. From the perspective of the Job Demands–Resources (JD-R) model, these extended hours represent intensified job demands that exceed available recovery and support resources, resulting in progressive strain and energy depletion. These effects span multiple domains, including physical and psychological exhaustion, burnout, diminished motivation, and work–life imbalance. In Latvia, where long shifts are still widely used, these risks are especially salient and often not adequately addressed.

Physical strain arises from the cumulative exposure to demanding physical tasks and constant vigilance. Healthcare professionals spend long hours standing, moving patients, performing technical procedures, and maintaining continuous focus and responsibility. Under a 24 h arrangement, this physical burden becomes prolonged and intensified, leading to fatigue accumulation, musculoskeletal disorders, and a reduced ability to respond effectively to unexpected clinical situations [45,46,47]. Within the Effort–Recovery theory, such insufficient recovery opportunities hinder the body’s physiological restoration processes, predisposing employees to chronic fatigue and performance decline.

Psychological strain is even more pronounced during long shifts. Continuous exposure to emotionally intense experiences, such as patient suffering, end-of-life care, ethical dilemmas, and complex team communication, exceeds emotional regulation capacity [41,42,43,48]. When sustained effort is not followed by adequate recovery, emotional exhaustion and depersonalisation occur, aligning with the Effort–Recovery and JD-R frameworks. These conditions are the hallmarks of burnout syndrome, which, according to the WHO definition, arises from chronic workplace stress that is unsuccessfully managed, leading to emotional exhaustion, cynicism, and reduced professional efficacy [41,48]. National data from the Centre for Disease Prevention and Control (SPKC) and Riga Stradiņš University (RSU) show that 50–60% of nurses experience burnout symptoms, particularly under long-shift conditions and high workload [49,50,51]. This pattern reflects an organisation-level failure to balance job demands with adequate resources, as conceptualised in the JD-R model.

Extended shifts also impair cognitive functioning and decision-making capacity. Studies show that cognitive decline begins after 8 h of continuous work, becomes pronounced after 12 h, and even critical after 16–24 h [52,53,54]. Studies by the American Nurses Association (ANA) and other international studies show that the error rate increases by 200% after 12 h shifts compared with 8 h shifts [55,56].

Quality control reports in Latvian hospitals also note that the frequency of errors and incidents increases at the end of shifts, when staff are most tired. These errors include medication administration errors, documentation errors, delayed decisions, and misinterpretation of the patient’s condition [57,58]. However, many errors are not recorded, especially so-called ‘silent errors’, which occur more frequently during night shifts and at the end of shifts. Cognitively, this process exemplifies the “resource depletion” pathway within the JD-R framework, whereby sustained attentional effort without adequate recovery diminishes accuracy and increases risk.

Work–life imbalance is another major problem. During 24 h shifts, employees are often unable to rest properly, recover, or participate in family and social life. This particularly affects women, who are more likely to take on responsibility for the household and children [56,57,58,59,60]. In the long term, this imbalance contributes to social isolation, relationship difficulties, depression, and reduced motivation to work in healthcare. The Work–Life Interface framework explains these outcomes as the spill-over of excessive job demands into personal domains, leading to work–family conflict and diminished life satisfaction.

Eurofound studies have found that employees who experience chronic work–life imbalances are less likely to engage in professional development, change jobs more frequently, and leave the profession altogether [61]. This trend is confirmed in Latvia, with 29% of nurses in an RSU study admitting a desire to leave healthcare within the next two years [51].

Extended work patterns also limit opportunities for continuous learning and team integration. Due to fatigue and scheduling constraints, employees often cannot attend training, conferences, or supervision sessions, reducing opportunities for skill development and professional growth [62,63]. Fragmented schedules hinder teamwork, as staff from different shifts rarely meet, leading to communication gaps and reduced continuity of care [64]. From a systemic standpoint, these factors erode the social and professional resources that buffer against stress, intensifying the JD-R cycle of high demand and low support.

In summary, prolonged shifts create a self-reinforcing cycle of excessive demands, inadequate recovery, and disrupted work–life integration. The resulting burnout, cognitive decline, and reduced teamwork capacity pose not only individual risks but also systemic threats to patient safety and workforce retention. These outcomes directly align with the predictions of the Job Demands–Resources, Effort–Recovery, and Work–Life Interface frameworks, illustrating that unsustainable scheduling practices constitute a structural determinant of both professional health and care quality. Policy reforms aimed at balancing demands, improving recovery opportunities, and supporting professional development are therefore essential to ensure a sustainable healthcare workforce in Latvia.

### 4.3. Patient Safety Indicators

Patient safety is one of the key quality indicators of a healthcare system and is directly affected by the ability of healthcare staff to perform clinical activities in an accurate, timely manner and high quality. Within the Job Demands–Resources (JD-R) framework, patient safety outcomes can be interpreted as downstream indicators of system imbalance, where excessive job demands and insufficient recovery opportunities reduce staff vigilance, decision-making quality, and capacity for safe care delivery. Long work shifts, especially 24 h shifts, significantly impair this ability by causing chronic fatigue, impaired concentration, reduced reaction speed, and increased likelihood of cognitive errors [65,66,67,68].

Staff fatigue is considered one of the most important risks to patient safety. International studies show that after 17 h of continuous wakefulness, an employee’s reaction time and decision-making ability are comparable to those observed at a blood alcohol concentration of 0.05% [67]. From the perspective of the Effort–Recovery theory, such extended wakefulness reflects an absence of restorative processes, leaving the cognitive system functionally depleted. This indicates that staff deprived of rest may function at an impairment level comparable to mild intoxication, posing clear risks in settings where clinical decisions have direct consequences for life and safety.

Studies also confirm that the frequency of errors increases significantly after 12 h of work, and errors are much more frequent at night and at the end of shifts [13,57,58,69]. The most common categories of error include incorrect medication dosage or timing, documentation inaccuracies, patient identification errors, delayed responses to clinical changes, and misjudged decisions in critical care situations [58]. These outcomes are consistent with the cognitive fatigue mechanisms described in the JD-R and Effort–Recovery models, where sustained effort without adequate recovery results in diminished attentional control and working-memory overload.

Latvian medical treatment institutions often lack systematic error analysis, as many errors are not recorded, especially so-called ‘silent errors’ in documentation or patient assessment. Such errors are rarely investigated and originate from sources such as fatigue-related inattention and are not documented, thus preventing evidence-based improvements [4,58].

International studies indicate that in hospitals where 12+ hour shifts are regularly used, patients are at higher risk of medication errors, infections, delayed treatment, and other adverse outcomes. For example, a 2014 Health Affairs study found that patient mortality was higher in hospitals where staff worked longer shifts [70]. A 2020 BMJ Quality & Safety study analysing more than 200 hospitals in Europe and the US confirmed that long shifts significantly worsen patient outcomes [65]. These findings support the theoretical argument that organisational structures generating excessive demands without compensatory recovery periods translate into measurable declines in system-level safety performance.

In addition to objective clinical indicators, patients’ subjective assessment of the quality of care is also important. Studies show that patient satisfaction decreases when staff are overworked or fatigued [41,71,72]. Fatigue reduces attentiveness, empathy, and communication quality, which patients interpret as indifference or neglect. Over time, such perceptions undermine trust, increase complaints, and damage institutional reputation. This aligns with the Work–Life Interface framework, where occupational fatigue and emotional exhaustion spill over into interpersonal interactions, eroding the perceived quality of care.

Both the WHO and the International Labour Organization (ILO) have long recognised that patient safety is inseparable from staff working conditions [65,70]. WHO guidelines explicitly recommend limiting shifts to 12 h, ensuring at least 11 consecutive hours of rest between shifts, and providing systematic education on fatigue risks and error prevention. Similarly, EU Directive 2003/88/EC—which is binding in Latvia—sets out the right to weekly rest and limits total weekly working hours, including overtime, to 48 h [1]. However, these principles are not consistently enforced in Latvian healthcare due to persistent staff shortages, limited digital scheduling infrastructure, and entrenched workplace traditions.

EU Directive 2003/88/EC concerning certain aspects of the organisation of working time, which is also binding in Latvia, establishes the right to weekly rest and a maximum weekly working time of 48 h, including overtime [1]. However, these norms are often not consistently applied in healthcare in Latvia for practical reasons such as staff shortage, incomplete IT systems, or traditions.

In addition, studies show that in countries where the transition from 24 h shifts to a structured three-shift system has already been implemented, patient safety is significantly improved. Experience in Sweden shows that the number of errors in care decreased by 30% in the three years following the transition to 8 h shifts [19]. Similar results have been observed in Norway and Germany, where long shifts are banned or strictly regulated [18].

Overall, the quality of working time management constitutes a structural determinant of patient safety. Prolonged and unregulated shifts create cumulative fatigue, delay critical decisions, and increase preventable errors, thereby representing a systemic risk to both staff well-being and healthcare quality. Within the JD-R and Effort–Recovery frameworks, this relationship is cyclical and resource depletion leads to fatigue, which compromises performance, further intensifying job demands. Breaking this cycle requires policy-level intervention and institutional commitment to rest regulation, staffing adequacy, and digitalised monitoring.

These principles can be operationalised through a phased transition pathway: preparation → pilot projects → evaluation → scale-up → continuous monitoring—as illustrated in Figure 1. This stepwise approach allows testing of scheduling reforms in controlled environments before nationwide adoption, ensuring that safety improvements are evidence-based and sustainable.

To contextualise Latvia’s position, the following section provides a comparative overview of international shift models and their implementation outcomes. These examples demonstrate how alignment with occupational health theories and policy frameworks can effectively transform working conditions, strengthen workforce resilience, and enhance patient safety.

### 4.4. International Policies and Models

International experience consistently demonstrates that 24 h shifts in healthcare pose substantial risks to both employees and patients. In line with the Job Demands–Resources (JD-R) framework, such extended schedules reflect structural over-demand and insufficient recovery resources, creating conditions that undermine both staff health and patient safety. Consequently, many countries have restructured their working time organisation, transitioning to safer, evidence-based models that balance flexibility with regulatory safeguards. Three-shift systems with 8 or 12 h shifts and modular, digitally supported scheduling now prevail across much of Europe, North America, and Oceania [6,15,18,19,20,21,22,23,24,25,26,73].

Nordic countries such as Norway, Sweden, Denmark, and Finland are pioneers in introducing flexible, safe schedules that are more conducive to employee well-being. 24 h shifts are no longer used in these countries. Three-shift systems with 8 h shifts or flexible 12 h schedules are predominant, closely monitored and structured to ensure at least 11 h of rest between shifts [18,19,20]. These arrangements directly operationalise the Effort–Recovery principle, ensuring sufficient recovery time restores cognitive and emotional resources, which empirical data confirm through reduced burnout and higher patient satisfaction.

Germany mainly uses shift systems of 8 to 12 h. Since 2003, the legislation in force clearly limits the length of a shift and requires mandatory rest between shifts [18,19,20,21]. In addition, strict overtime accounting and compensation mechanisms are in place. The German example shows how regulatory support can achieve a sustainable shift system.

In France, most hospitals have switched to a 35 h week with 7–8 h working days. Longer shifts are only allowed in emergency situations and are carefully monitored [18,22]. In France, the focus is on staff well-being, safety culture, and continuity of care.

In the United Kingdom (UK), both classic 8 h and 12 h rotating shifts are used, but work schedules are often based on collective agreements with employees [18,23]. The UK is also trialling split shift models, where a shift is split in two parts, for example with a longer break in the middle of the day, allowing flexibility in staff workload.

In Canada, 12 h shifts are prevalent, but their application is closely linked to ensuring adequate rest. In many cases, mixed schedules are used, combining 8 h and 12 h shifts to suit the needs of the units and staff availability [24]. This hybrid scheduling demonstrates that flexibility, when underpinned by recovery guarantees, can preserve efficiency without compromising safety.

In Australia and New Zealand, 8, 10, and 12 h shifts are used, depending on the care sector. Compressed working weeks are also becoming popular, such as four 10 h shifts per week, which give more days off but require a high degree of self-discipline and planning [25].

In the USA, 12 h shifts have historically dominated, but more and more hospitals are moving towards more flexible and dynamic schedules, using technological solutions and data analytics [26]. For example, healthcare organisation Kaiser Permanente has transitioned to hybrid shift models that combine safety, employee choice, and efficiency.

International evidence shows that the transition from a 24 h to a three-shift system has several benefits:Up to 30% reduction in burnout [19,20];25–35%. reduction in error rate [70,73];Increased patient satisfaction and a more positive perception of the quality of care [41];Reduced staff turnover and greater retention in the sector [14].

In Sweden, within three years of completely abandoning 24 h shifts, the number of recorded errors in care decreased by 30%, the number of sick leaves by 20%, and employee satisfaction with work increased by 35% [19]. These outcomes validate the theoretical premise that structured rest and predictable scheduling act as key “resources” within the JD-R model, mitigating chronic fatigue and fostering long-term engagement. The Nordic models are particularly notable for the role of social partnership and collective agreements, which ensure that work schedules are based not only on patient needs but also on considerations of employee well-being. This practice promotes long-term trust in the system and employee loyalty.

As summarised in Table 2, most OECD and EU countries have already introduced shorter and better-regulated work schedules. The table compares standard shift durations, rest requirements, implementation strategies, and key outcomes across selected health systems. These examples demonstrate how legal, managerial, and cultural adjustments can produce measurable gains in staff safety, job satisfaction, and retention, while maintaining efficiency in service delivery.

Taken together, these comparative findings highlight that sustainable reform depends on regulatory clarity, consistent rest enforcement, and active social partnership between policymakers and healthcare professionals. Latvia’s continued reliance on 24 h shifts underscores the urgency of aligning national legislation and practice with EU and WHO occupational safety standards. Integrating the JD-R and Effort–Recovery frameworks into this process would allow Latvia to design policies that explicitly balance job demands with recovery resources, transforming working time regulation from an administrative adjustment into an evidence-based health protection measure.

The international evidence highlights viable pathways for reform. The discussion that follows integrates these insights with the theoretical models outlined earlier, translating them into practical policy options for Latvia.

## 5. Discussion

### 5.1. Possible Alternative Shift Models in Latvia

The transition from 24 h shifts to safer and more sustainable work schedules requires not only political will and regulatory solutions, but also a clear understanding of practically feasible models adapted to the specifics of the Latvian healthcare system. In line with the Job Demands–Resources (JD-R) and Effort–Recovery frameworks, the reform should aim to optimise the balance between workload intensity and recovery opportunities, ensuring that organisational structures actively protect employee well-being. International experience shows that several alternative systems can be applied depending on the profile of the unit, the availability of resources, and the staff structure [18,19,20,21,22,23,24,25,26]. Below are four possible shift models that could replace 24 h shifts in Latvia.

1.
*12 h shift model (day/night/rest)*


This is the most frequently used transitional system after discontinuing 24 h shifts. A typical rotation follows a three-day cycle: day shift, night shift, and rest. The model is relatively easy to implement and ensures predictable rest periods. Advantages include simple planning and continuity, while *challenges* include persistent fatigue after night duties and higher staffing requirements to cover all hours of care. From a JD-R perspective, this model moderately reduces excessive demands but still leaves limited recovery capacity, requiring careful monitoring of fatigue accumulation.

2.
*Maximum shift duration—16 h or less*


By legislatively setting a 16 h limit, organisations can reduce extreme workloads while preserving flexibility. This configuration serves as a transitional stage between 24 h and 12 h systems. Although it reduces acute overload, it remains suboptimal from a health perspective, since fatigue and cognitive decline become significant after 12 h of work. Within the Effort–Recovery model, this approach partially mitigates strain but does not prevent cumulative fatigue, providing only an interim improvement rather than a sustainable solution.

3.
*Modular work (2 × 6 h or 3 × 8 h shifts per day)*


The modular system provides for an equal distribution of hours per day across shifts. Popular examples:6 h shifts: 07:00–13:00, 13:00–19:00, 19:00–01:00, 01:00–07:00;8 h shifts: 07:00–15:00, 15:00–23:00, 23:00–07:00.

This structure enhances communication quality, reduces cognitive fatigue, and improves continuity of care. While administratively complex and resource-intensive, it best aligns with the Effort–Recovery principle by providing frequent restorative intervals and predictable cycles.

4.
*Flexible shifts according to the unit profile*


Flexible models adapt shift lengths to unit demands and individual circumstances—for instance, retaining 12 h shifts in intensive care but introducing 8 h schedules in chronic care settings. This approach allows personalisation based on caregiving responsibilities, education, or health needs. According to the Work–Life Interface framework, such flexibility enhances employee autonomy, life balance, and retention, which are key determinants of sustained motivation and reduced burnout.

These models (see Table 3) are not mutually exclusive These models are not mutually exclusive. Rather, they can be combined according to unit-specific needs, staff composition, and institutional strategy. The overarching policy goal is to replace 24 h scheduling with structured, health-oriented alternatives that promote sustained staff engagement and patient safety.

Pilot testing these models across diverse institutions will allow for evaluation, refinement, and gradual expansion. This evidence-informed, iterative approach reflects the JD-R logic of adjusting organisational resources to changing demands, thereby reducing resistance and enhancing implementation feasibility.

### 5.2. Implementation Prerequisites & Feasibility

The transition from 24 h to structured shift systems represents a systemic reform that extends beyond technical scheduling—it requires cultural transformation within the healthcare environment. The process must operationalise the JD-R and Effort–Recovery principles: reducing excessive job demands while strengthening recovery mechanisms through regulation, staffing, and technology.

Firstly, the regulatory framework requires review and adaptation. To provide a legal basis for the introduction of new shift models, the Labour Law should be amended to set a maximum duration per shift, e.g., 12 or 16 h, and specific provisions should be included on the organisation of work in healthcare, taking into account its specific nature. Collective agreements should play a greater role in shaping individual agreements and local solutions, while the Ministry of Health is recommended develop guidelines on recommended work schedules, their organisation, and monitoring mechanisms. These documents should be drawn up in accordance with the European Union Directive 2003/88/EC concerning certain aspects of the organisation of working time, ensuring consistency with international occupational safety standards [18,20,21,22,23,24,25,26].

Secondly, provision of human resources is a prerequisite for a successful transition. The combination of higher numbers and shorter shift lengths means additional staff units are needed. This is especially true for nights, weekends, and intensive care units, where the intensity of care is higher. Job descriptions should clearly define shift durations, reassignment of duties, and handover procedures. At the same time, incentive mechanisms could be developed to encourage nurses and other healthcare specialists to adopt flexible schedules, such as night differentials, bonuses for flexible availability, or opportunities to work part-time with a reasonable workload distribution. A strategy should also be developed for returning employees to the labour market from parental leave or after a period of burnout.

Thirdly, financing is essential. The new shift system means higher expenses for remuneration, information systems, and transition support. Therefore, it is necessary to ensure additional funding from the state budget at least for the transition period, for example, three years, as well as to provide earmarked grants to medical treatment institutions for the implementation of pilot projects. It is important to consider the possibility of using the European Union Structural Funds or the Recovery and Resilience Mechanism, as well as reviewing tariffs in certain areas of care to ensure long-term sustainability.

Fourthly, targeted management and staff education are critical components of the transition process. Changing schedules is not just a technical task—it requires changes in management skills, approaches to workforce planning, and communication with employees. Training should therefore be provided to heads of units, chief nurses, and HR specialists on dynamic and balanced work scheduling. At the same time, staff participation in decision-making should be strengthened, for example through internal forums, surveys, or pilot schedules to build trust and reduce resistance. An additional aspect involves clarifying widespread misconceptions about the ‘effectiveness’ of 24 h shifts, using evidence on error rates, burnout, and patient safety.

Fifthly, technological infrastructure is an essential prerequisite for the introduction of flexible work schedules. In modern practice, successful personnel management is unthinkable without specialised software that allows planning shifts, recording workload, monitoring rest periods, and ensuring transparency for all parties involved. Such a system is not yet widespread in Latvia, but its introduction would be a strategic investment in the quality of healthcare and staff retention. Consideration could be given to introducing a centralised solution for all public authorities, which would ensure both uniform accounting and transferable practices between different profiles.

Sixthly, it is advisable to introduce pilot projects in different types of medical treatment institutions, such as a university hospital, a regional hospital, a psychiatric ward, or an emergency medical service, before a general reform. These pilot projects would allow different shift models and their impact on workload, patient safety, staff satisfaction, and financial costs to be tested in practice. Only after such a practical evaluation would it be justified to decide on general introduction at national level. Pilot projects should follow a clear framework with defined objectives, indicators, and evaluation mechanisms.

Finally, continuous monitoring and evaluation of results is needed. Monitoring indicators need to be established, such as compliance of work schedules with norms, number of employee sick leaves, burnout level dynamics, patient error statistics. The Ministry of Health could consider gradually phasing out 24 h shifts, establish a coordinating body responsible for implementing, evaluating and, where necessary, adjusting this reform. Annual progress reports can serve as a tool for both public information and policy adjustments.

Together, these prerequisites constitute a framework that translates theoretical constructs into practical policy mechanisms. By embedding recovery, participation, and regulation into workforce management, Latvia can achieve a systemic rebalancing of job demands and resources and protecting both staff health and patient safety.

#### Implementation Challenges and Mitigation Strategies

Despite broad support for transitioning to safer shift systems, several implementation challenges remain.

Staff shortages and workload pressure: Latvia’s healthcare system faces a chronic shortage of nurses, making it difficult to ensure adequate staffing during shorter shifts. *Mitigation:* Phased introduction of 12 h or 8 h models with pilot projects and flexible staffing pools.Financial implications: Adjusting shift schedules may temporarily increase costs due to overtime compensation and recruitment needs. *Mitigation:* Align reform with EU funding streams and workforce development programs (2025–2029 strategy).Cultural and organizational resistance: Long-standing traditions of 24 h shifts create psychological and managerial inertia. *Mitigation:* Education of nurse managers, clear communication of safety benefits, and inclusion of staff in schedule design.Monitoring and sustainability: Without a strong data infrastructure, early reforms risk losing momentum. *Mitigation:* Establish a national dashboard (as in Figure 1) to track compliance, rest, and patient safety indicators.

Each of these barriers corresponds to a misalignment within the JD-R framework, excessive demands, inadequate resources, or unclear feedback loops. The proposed mitigation strategies aim to restore this balance through systemic, data-informed management.

### 5.3. Barriers and Enabling Factors for Implementation

While the rationale for reform is strong, several structural and cultural barriers may hinder progress, the most notably workforce shortages, financial constraints, and normalised overwork. At the same time, enabling factors such as transparent communication, participatory planning, and targeted state funding can facilitate acceptance. International evidence shows that visible government leadership, clear legal norms, and continuous evaluation are decisive in maintaining reform momentum.

From a theoretical standpoint, these enablers represent the transformation of *resources* within the JD-R model, from individual coping to institutional support. Through this shift, Latvia can move from a reactive system that relies on resilience to a proactive one that systematically prevents burnout and ensures recovery capacity.

Ultimately, the transition from 24 h shifts to structured, flexible scheduling is not merely an administrative reform but a value-driven transformation of the healthcare work culture. Its success depends on recognising nurses and other professionals not as expendable labour resources, but as the core of a sustainable health system. By aligning national policies with occupational health theory and international standards, Latvia can create a future-oriented, evidence-based framework for workforce well-being and patient safety (see Figure 1).

## 6. Conclusions

The evidence presented in this study underscores the urgent need for Latvia to transition from 24 h shifts to structured, health-oriented scheduling models. Such a transformation is not merely an administrative adjustment but a systemic reform grounded in occupational health theory, ethical responsibility, and evidence-based governance. A gradual, phased implementation, beginning with pilot projects in high-workload units, offers the most feasible pathway for reform. These pilots would enable the evaluation of multiple parameters, including staff workload, error rates, sick leaves, and patient safety outcomes. The results would provide the empirical basis for selecting the most appropriate scheduling model for each clinical context, ensuring that decisions are informed by real-world data rather than normative assumptions.

National guidelines could specify acceptable shift lengths, minimum rest standards, and recommended schedule archetypes, co-developed with professional bodies and providers, including monitoring provisions. These guidelines must be developed in collaboration with professional organisations and medical treatment institutions themselves, ensuring compliance with both laws and regulations and the reality of care. The guidelines should also include mechanisms to monitor the impact of the implemented schedules on staff and patients.

Thirdly, additional investment in human resources is needed, in terms of numbers, quality, and motivation. A new schedule system requires more staff, but also more efficient use of them, such as more accurate shift planning according to the intensity of care, involving technical staff in certain processes, and stronger interprofessional cooperation. At the same time, employee well-being policies must be strengthened, psychological support provided, burnout reduced, and a sustainable professional identity developed.

Fourthly, we should enable the development of a unified, nationally integrated information system that allows for the effective management and analysis of staff schedules, workloads, shift lengths, and rest periods. Such a system would be an instrument not only for management, but also for policy makers, providing a data-driven basis for decision-making. This system should be designed to export, compare, and evaluate the practices of different institutions, facilitating the exchange of experience and the adaptation of innovations.

Fifthly, a broad public discussion may help reframe narratives around the working conditions of healthcare professionals, especially the public perception of the ‘culture of sacrifice’ associated with 24 h shifts. Professional burnout, errors, emotional stress, and disruption of personal life should not be romanticised as ‘a heroic deed’, but rather recognised as a systemic problem that requires a structured solution. That is why communication strategy is part of the change process—it helps reduce internal resistance, strengthen understanding, and generate long-term support.

Taken together, the analysis supports several key conclusions. The continued prevalence of 24 h shifts in Latvian healthcare is incompatible with international standards of patient safety and staff well-being. Empirical evidence confirms that prolonged shifts exacerbate fatigue, increase clinical errors, and accelerate professional burnout and turnover. In contrast, structured alternatives such as 12 h, 8 h, or flexible modular systems produce measurably better outcomes for both staff and patients. However, implementing these systems requires systemic readiness: legislative adaptation, financial support, managerial competence, technological infrastructure, and active engagement of healthcare professionals.

This transformation should therefore not be seen as a technocratic experiment, but as a necessary part of the reform that is building a human-centred, excellence-based healthcare system in Latvia. Working with people should not be based on selflessness as the main mechanism—it should be based on competence, balance, and respect.

These findings are consistent with the Job Demands–Resources (JD-R) and Effort–Recovery theoretical frameworks, which emphasize that reducing excessive job demands and ensuring sufficient recovery opportunities are essential for maintaining staff well-being and patient safety. Embedding these principles in shift reform ensures that organizational changes are grounded in evidence-based occupational health theory.

In essence, this transition represents a paradigm shift, from endurance-based to balance-based healthcare. It positions Latvia on a trajectory toward a human-centred, evidence-informed, and excellence-driven system in which staff well-being is viewed not as a secondary concern but as a prerequisite for patient safety and quality care.

## Figures and Tables

**Figure 1 ijerph-22-01736-f001:**
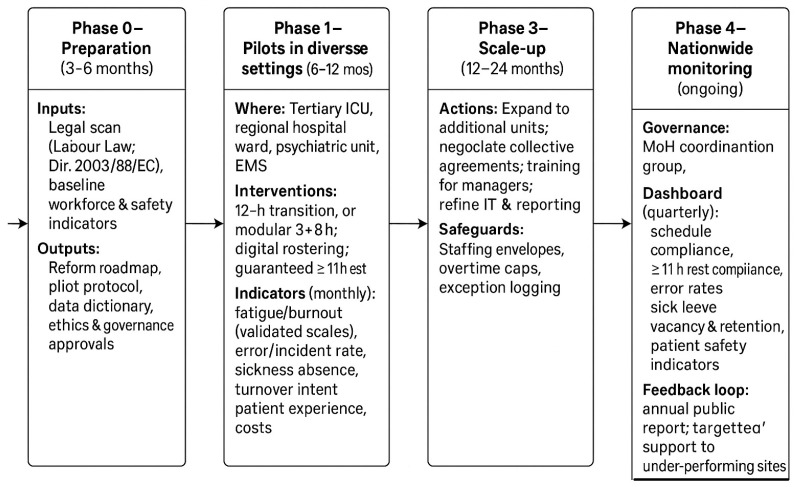
Phased transition framework from 24 h shifts to safer schedules in Latvian healthcare (pilot → evaluation → scale-up → nationwide monitoring).

**Table 1 ijerph-22-01736-t001:** Latvia vs. selected OECD/EU countries: nurses per 1000 population, proportion of >12 h shifts, burnout prevalence, and patient safety outcomes.

Indicator	Latvia	EU Average	Nordic Average
Nurses per 1000 population [10]	4.2	8.5	10.1
Share of nurses working >12 h [11,18]	60%	28%	10%
Reported burnout (moderate–high) [16,37]	57%	38%	25%
Reported clinical errors (%) [16,17,27,28]	18%	9%	6%
Legal cap for shift duration [18]	None	≤12 h	≤12 h

Note: Latvia remains below EU nurse density averages and lacks statutory limits on shift duration.

**Table 2 ijerph-22-01736-t002:** Overview of international scheduling models, rest requirements, implementation strategies, and reported outcomes [18,19,20,21,22,23,24,25,26,27,28,29,73].

Country/Region	Standard Shift Duration	Rest Requirements	Implementation Strategy	Reported Effects (Examples)
Sweden	8–12 h	≥11 h between shifts	National labour agreements; strong unit-level co-design	↓ burnout; ↑ retention; ↓ errors
Norway	≤12 h	≥11 h; weekly rest	Pilots in university hospitals → wider rollout	↑ satisfaction; ↓ sick leave
Denmark/Finland	8–12 h	EU/ILO aligned	Collective agreements; robust rostering IT	Better continuity; stable costs
Germany	8–10 (12) h	Mandatory daily/weekly rest	Working Time Act; strict overtime accounting	↓ med errors; improved compliance
France	7–8 h (35 h week)	Mandated rest/limits	Policy + hospital protocols; exceptions audited	↑ predictability; mixed on 12 h
United Kingdom	8 h & 12 h (rotations)	Mandated rest	Local agreements; split-shift pilots	Mixed; staff preference matters
Canada	12 h common; mixed 8/12 h	Mandated rest	Local policies; fatigue training	↓ incidents with robust rest
Australia/New Zealand	8/10/12 h	Mandated rest	Compressed weeks in some units	↑ days off; needs discipline
United States	12 h prevalent; hybrids	Rest varies by state	Large systems adopt hybrid models	↑ efficiency when coupled with rest
Latvia (current)	up to 24 h	No statutory ≤12 h cap	Historical practice; shortages	↑ fatigue; safety concerns

Note. ↑: decrease; ↓: increase. Countries implementing ≤12 h limits combined with structured rest demonstrate consistently lower burnout and error rates. Latvia remains an outlier in lacking formal regulation of shift duration and rest compliance.

**Table 3 ijerph-22-01736-t003:** Comparative Table on Alternative Shift Models [18,19,20,21,22,23,24,25,26,27].

Model	Satisfaction (%)	Patient Safety (%)	Efficiency (%)	Notes
12 h shift model	75	78	80	Available in units with lots of resources
Max. 16 h shifts	68	72	74	Transitional solution
Modular (2 × 6 h or 3 × 8 h)	82	88	86	High level of coordination required
Flexible shifts	90	93	92	Staff planning platform required

## Data Availability

The data presented in this study are available on request from the corresponding author. The data are not publicly available due to ethical and privacy restrictions in accordance with the approval of the Riga Stradiņš University Ethics Committee (protocol code 2-PĒK-4/416/2023, 5 September 2023).

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
