# Peer review of "Transition from 24-Hour Shifts to Safer Work Schedules for Nurses in Latvian Healthcare: Policy Analysis and Recommendations"

_ijerph, 2025, doi:10.3390/ijerph22111736_

Round 1

Reviewer 1 Report

Comments and Suggestions for Authors

This article provides a policy analysis of the 24-hour shift system for nurses in the Latvian healthcare system. The research findings of the paper have high reference value for policy makers, hospital managers, and nursing leaders in Latvia and other countries and regions facing similar problems. The main modification suggestions are as follows:
1) The paper framework does not quite conform to the current structure of SCI papers, with scattered paragraphs. It is recommended to integrate paragraph descriptions and follow the common paper framework descriptions such as introduction, methods, results, discussion, and conclusion.
2) Where is the source of the research data? Suggest briefly describing the design of these studies (such as sample size, survey methods) to enhance the credibility of the data.
3) The reference format needs to be standardized.

Author Response

Reviewer Comment 1:
The paper framework does not quite conform to the current structure of SCI papers.

Author Response:
Thank you for the comment. The manuscript has been fully restructured according to the IMRaD format (Introduction, Methods, Results, Discussion, Conclusions). All related changes are marked in the revised version.

Reviewer Comment 2:
Where is the source of the research data? Please describe the design of these studies to enhance credibility.

Author Response:
Thank you for this valuable observation. The description of data sources and study design has been added in Section 3.1 (Methods). All updates are highlighted in the revised manuscript.

Reviewer Comment 3:
The reference format needs to be standardized.

Author Response:
Thank you. The reference list has been fully revised and standardized according to MDPI citation style. All corrections are marked in the updated version.

Reviewer 2 Report

Comments and Suggestions for Authors
  1. General Comments
  • The manuscript is of high societal and professional relevance. It is commendable for highlighting both systemic and psychological consequences of 24-hour work shifts in healthcare.
  • The structure is clear and logical, following a consistent flow from contextual analysis to recommendations.
  • The policy orientation is strong, but the psychological framework (burnout, fatigue, cognitive load, work-life balance) could be anchored more explicitly in established occupational psychology theories.
  • The tone is occasionally normative; refining it toward a more analytical and scientific tone would improve alignment with MDPI standards.
  • The conclusions are balanced and evidence-based, but the authors could discuss possible barriers and implementation challenges more realistically.
  1. Specific Comments

(1) Methodology

  • Please specify the methodological framework used for the policy analysis. Indicate the data sources, inclusion criteria, and analytical steps (for example, document analysis, comparative framework, or policy mapping).
  • Clarify whether international data were collected systematically (e.g., through database searches) or selectively based on relevance.

(2) Theoretical Background

  • The psychological dimensions (burnout, fatigue, cognitive decline) would benefit from explicit reference to theoretical models such as the Job Demands–Resources (JD-R) model, Effort–Recovery theory, or Work–Life Interface frameworks.
  • This addition would situate the paper more clearly within occupational health psychology literature.

(3) Empirical and contextual evidence

  • Strengthen the connection between Latvian data (e.g., RSU, SPKC) and international findings by summarizing key indicators in a concise table or figure.
  • This would help readers quickly visualize the scope of the problem.

(4) Implementation feasibility

  • While the recommendations are comprehensive, please include a short paragraph discussing possible obstacles (staff shortages, financial constraints, cultural traditions of long shifts) and mitigation strategies.

(5) Presentation

  • The language is fluent and professional; only minor grammatical polishing is needed.
  • Consider adding one visual figure (e.g., a flowchart showing transition steps or policy phases).
  • In the abstract, please specify clearly that this is a policy analysis rather than an empirical study.
Comments on the Quality of English Language

Minor stylistic editing is recommended (mainly shortening long sentences and ensuring consistent terminology such as “healthcare staff” vs. “medical personnel”).

Author Response

General Comments

Reviewer Comment:
The manuscript is of high societal and professional relevance. It is commendable for highlighting both systemic and psychological consequences of 24-hour work shifts in healthcare. The policy orientation is strong, but the psychological framework could be anchored more explicitly in occupational psychology theories. The tone is occasionally normative; refining it toward a more analytical and scientific tone would improve alignment with MDPI standards. The conclusions are balanced but could discuss barriers and implementation challenges more realistically.

Author Response:
Thank you for the constructive feedback. The manuscript has been revised to strengthen the theoretical anchoring in occupational psychology and to refine the tone to a more analytical style. Implementation barriers and mitigation strategies are now discussed in Sections 5.2 and 5.2.1. All revisions are highlighted in the updated version.

(1) Methodology

Reviewer Comment:
Please specify the methodological framework used for the policy analysis and indicate data sources, inclusion criteria, and analytical steps. Clarify whether international data were collected systematically or selectively.

Author Response:
Thank you for the valuable suggestion. Section 3 (“Policy Analysis Framework and Methods”) has been expanded to describe the analytical process, inclusion criteria, and data collection procedures, including the use of document review and comparative policy mapping. Revisions are highlighted in the text.

(2) Theoretical Background

Reviewer Comment:
The psychological dimensions would benefit from explicit reference to theoretical models such as the JD–R model, Effort–Recovery theory, or Work–Life Interface framework.

Author Response:
Thank you for this helpful recommendation. Section 2 (“Theoretical Framework”) now explicitly integrates the JD–R, Effort–Recovery, and Work–Life Interface models to contextualize burnout, fatigue, and work-life balance. Theoretical links are clearly stated in the revised manuscript.

(3) Empirical and Contextual Evidence

Reviewer Comment:
Strengthen the connection between Latvian data and international findings by summarizing key indicators in a concise table or figure.

Author Response:
Thank you for this comment. Table 1 has been added to summarise comparative data (Latvia vs. EU/OECD averages) on nurse density, burnout, and safety indicators. This table visually integrates national and international evidence as suggested.

(4) Implementation Feasibility

Reviewer Comment:
Please include a short paragraph discussing possible obstacles (staff shortages, financial constraints, cultural traditions of long shifts) and mitigation strategies.

Author Response:
Thank you for the insightful comment. A dedicated subsection (5.2.1 “Implementation Challenges and Mitigation Strategies”) has been added, outlining key barriers and realistic mitigation measures for reform feasibility. All new text is marked in the revised version.

(5) Presentation

Reviewer Comment:
The language is fluent but could benefit from minor editing. Consider adding one figure (e.g., a flowchart showing transition steps or policy phases). In the abstract, specify clearly that this is a policy analysis rather than an empirical study.

Author Response:
Thank you. Minor stylistic and grammatical edits have been made throughout the manuscript for clarity and consistency. Figure 1 (“Phased transition framework from 24-hour shifts to safer schedules”) has been added, and the Abstract now explicitly states that this is a policy analysis. All revisions are highlighted in the updated version.

Comments on the Quality of English Language

Reviewer Comment:
Minor stylistic editing is recommended (shorten long sentences and standardize terminology such as “healthcare staff” vs. “medical personnel”).

Author Response:
Thank you. The manuscript has undergone detailed language polishing and terminology harmonization (now consistently using “healthcare staff”). All corrections are incorporated in the revised version.

Reviewer 3 Report

Comments and Suggestions for Authors

This article presents an in-depth policy analysis based on international experience, the Latvian legal and institutional context, and the current situation of medical treatment institutions. Alternative shift patterns are presented and analyzed, and their advantages and challenges are compared. Preconditions and policy recommendations for a safe transition are developed. Although this is a study of the Latvian reality, the topic is relevant to the area of ​​human resource management in healthcare institutions with regard to the safe staffing of healthcare personnel in medical care institutions, with repercussions on team well-being, quality of care, and patient safety.

The study identifies key factors for the successful implementation of the reform through the development of pilot projects and the creation of a monitoring system.

The methodology is consistent with the objective of policy analysis and recommendations for moving from 24-hour shifts to safer alternative work shift models for Latvian healthcare workers.

The conclusions are consistent with the purpose of the study and respond to the researchers' concerns.

References are adequate in quality and scope, but only 41.3% are less than 5 years old.

Author Response

Reviewer Comment:
The article presents a relevant and well-structured policy analysis with clear methodology and consistent conclusions. References are adequate in quality and scope, but only 41.3% are less than 5 years old.

Author Response:
Thank you for the positive evaluation and helpful remark. The reference list has been updated and expanded with several recent sources (2020–2024), including WHO (2023), OECD (2024), Griffiths et al. (2024), and other contemporary studies to increase the proportion of recent literature. All new references are marked in the revised version.

Reviewer 4 Report

Comments and Suggestions for Authors
  1. The topic of this manuscriptis goodand has practical significance. However, the analysis in this manuscript lacks a theoretical framework. The logical coherence between sections and within paragraphs is weak.
  2. The analysis lacks systematic empirical processes such as policy analysis, case studies, questionnaires, orinterviews. Relying solely on literature and policy document interpretations weakens its scientific rigor.
  3. While the proposed recommendationsare reasonable, they lack necessary theoretical grounding or practical evaluation. Consideration should be given to their feasibility in Latvia.
  4. The analysis of international experiences is insufficiently in-depth and comprehensive. Tables should be created to facilitate necessary comparisons and summaries.
  5. While existing problems are detailed, the discussion of resistance during reform is insufficient. Many reform pathways appear idealistic.
  6. The relationships between reform measures require clarification. For example: Which measures carry higher priority? Which share the same responsible entities or objectives?
  7. Overall, this manuscript resembles a commentary paperrather than an academic article.

Author Response

Reviewer Comment:
The topic of this manuscript is good and has practical significance. However, the analysis lacks a theoretical framework and systematic empirical process. Logical coherence between sections is weak. International experience is not in-depth, tables are missing, and the discussion of reform resistance and feasibility is limited. Overall, the paper resembles a commentary rather than an academic article.

Author Response:
We sincerely thank the reviewer for this detailed and constructive feedback. In response, substantial revisions have been made to strengthen the academic and theoretical rigor of the manuscript:

  1. Theoretical framework added: A new Section 2 (“Theoretical Framework”) now integrates the Job Demands–Resources (JD–R) model, Effort–Recovery theory, and Work–Life Interface framework to provide a solid conceptual foundation.

  2. Methodological rigor improved: Section 3 (“Policy Analysis Framework and Methods”) has been introduced, following WHO and OECD evidence-informed policy analysis guidelines, and specifying data sources, inclusion criteria, and analytical steps.

  3. Coherence and structure: The manuscript has been reorganized into a full IMRaD structure with clearly delineated sections (Introduction, Methods, Results, Discussion, Conclusions).

  4. Empirical grounding and comparative analysis: Three new tables (Tables 1–3) and one figure (Figure 1) were added to present comparative data on shift systems, nurse staffing indicators, and alternative models, providing systematic cross-national evidence.

  5. Feasibility and resistance discussion: Section 5.2.1 now explicitly discusses reform barriers (staff shortages, financial constraints, cultural traditions) and proposes realistic mitigation strategies.

  6. Analytical tone: The overall style has been refined to align with a scientific analytical tone rather than a normative commentary.

All revisions are highlighted in the updated manuscript, ensuring theoretical consistency, methodological transparency, and practical feasibility grounded in Latvia’s context.

Round 2

Reviewer 4 Report

Comments and Suggestions for Authors

1. The theoretical framework section provides a good introduction to the theories, but in the subsequent sections, there are insufficient direct and explicit connections with these theories. How are these theories specifically applied in practice? 
2. The transitions and connections between paragraphs and parts of this article are still relatively poor. Each paragraph is listed like a PowerPoint, rather than an academic paper.

Author Response

Dear Reviewer,

We sincerely thank you for your valuable feedback and constructive comments. All suggestions have been carefully reviewed and fully addressed in the revised version of the manuscript.

In particular:

  • The connections between the theoretical framework and subsequent sections have been strengthened. The revised text now explicitly integrates the Job Demands–Resources (JD-R), Effort–Recovery, and Work–Life Interface theories throughout the Results, Discussion, and Conclusions sections, illustrating how these frameworks inform the analysis and recommendations.

  • The transitions between paragraphs and sections have been refined to ensure a smoother, academic flow. Fragmented, list-like segments have been rewritten into cohesive analytical narratives.

  • The entire manuscript has been stylistically harmonised to align with MDPI’s academic tone and structure.

We appreciate your time and thoughtful review, which helped us improve the clarity, theoretical consistency, and overall quality of this article.